# Senescence-driven solubilization of biomass is the main source of kelp-derived dissolved organic carbon to the coastal ocean

Chance J. English [1] ✉, Tom W. Bell [2], Keri Opalk [1], David A. Siegel [3] & Craig A. Carlson [1]

Kelp forests form some of the most productive areas on earth and are proposed to sequester carbon in the ocean, largely in the form of released dissolved organic carbon (DOC). Here we investigate the role of environmental, seasonal and age-related physiological gradients on the partitioning of net primary production (NPP) into DOC by the canopy forming giant kelp (*Macrocystis pyrifera*). Rates of DOC production were strongly influenced by an age-related decline in physiological condition (i.e. senescence). During the mature stage of giant kelp development, DOC production was a small and constant fraction of NPP regardless of tissue nitrogen content or light intensity. When giant kelp entered its senescent phase, DOC production increased substantially and was uncoupled from NPP and light intensity. Compositional analysis of giant kelp-derived DOC showed that elevated DOC production during senescence was due to the solubilization of biomass carbon, rather than by direct exudation. We coupled our incubation and physiological experiments to a novel satellite-derived 20-year time series of giant kelp canopy biomass and physiology. Annual DOC production by giant kelp varied due to differences in standing biomass between years, but on average, 74% of the annual DOC production by giant kelp was due to senescence. This study suggests DOC may be a more important fate of macroalgal NPP than previously recognized.

Dissolved organic carbon (DOC) serves important ecological and biogeochemical roles in the ocean, including the structuring of microbial communities and the sequestration of carbon[1]. While there are many sources of DOC to the ocean, including phytoplankton exudation and river discharge, less is known about the contribution from coastal vegetated ecosystems (CVEs), including those dominated by mangroves, seagrasses, and macroalgae. In recent years, there have been efforts to constrain the flux of carbon from CVEs, aiming to integrate these ecosystems into estimates of marine carbon sequestration (i.e., blue carbon)[2–4]. These efforts are critical, as there is a growing movement to restore, conserve, and expand CVEs to enhance their ecosystem capacity as carbon sinks and sequester atmospheric $CO_2$[5–7].

Marine macroalgae form some of the most productive areas on earth and fix an estimated 1–3% of marine net primary production (NPP)[8,9]. Unlike other CVEs, macroalgae do not store carbon in the benthos, and

most of their fixed carbon is exported from their habitats as DOC or particulate detritus[10–12]. A synthesis of macroalgal NPP and export pathways found that naturally occurring macroalgal systems potentially sequester 173 (range = 61−268) Tg C yr$^{-1}$, of which 70% is in the form of DOC[12]. However, uncertainties in macroalgal biomass, NPP, and assumptions of macroalgal DOC production, remineralization rates, and export efficiencies call these estimates into question[13,14]. A major uncertainty in the production and fate of macroalgal NPP is the fraction that is partitioned into DOC, which is reported to range from <1 to 76%[3,11,15–21]; therefore, understanding the controls on DOC release rates by macroalgae is critical to their integration into blue carbon budgets. Environmental factors such as light intensity and nutrient availability are considered key regulators of DOC release by aquatic primary producers (see review of the overflow hypothesis[22] in ref. 1). However, studies of macroalgae DOC release that only consider these two factors have reported conflicting results[11,19,21,23], suggesting that factors other

[1]Marine Science Institute/Department of Ecology, Evolution and Marine Biology, University of California Santa Barbara, Santa Barbara, CA, USA. [2]Department of Applied Ocean Physics and Engineering, Woods Hole Oceanographic Institution, Woods Hole, MA, USA. [3]Earth Research Institute/Department of Geography, University of California, Santa Barbara, CA, USA. ✉e-mail: cje@ucsb.edu

than extrinsic ones may regulate macroalgal DOC production. Unlike extrinsic factors, such as light and nutrient availability, little attention has been given to the intrinsic factors associated with macroalgae physiology and life cycles, such as senescence. Knowledge about physiology is critical as primary producers can undergo rapid physiological changes that modulate their response to environmental factors and impact biogeochemistry[24,25]. Therefore, we hypothesized that consideration of intrinsic (age, senescence) as well as extrinsic (light and nutrients) factors must be considered to improve our understanding of DOC production by macroalgae.

*Macrocystis pyrifera*, hereafter referred to as giant kelp, is a globally distributed species that forms canopies visible from space[26]. Single "plants" consist of up to hundreds of fronds, each with an average lifespan of about 100–120 days[27]. Each frond consists of a single stipe with leaf-like blades that photosynthesize and take up nutrients from the surrounding seawater. Growth occurs year-round through the initiation of new fronds, and tissue physiology, including its carbon to nitrogen ratio and chlorophyll *a* content, is influenced by the availability of light and nutrients[28]. As fronds grow, blades emerge from the growing tip, creating a gradient in blade age along the frond. This pattern of growth results in large age-distributions of giant kelp biomass both within and between individual plants[29]. As a consequence of age, and regardless of ambient environmental conditions, giant kelp undergoes progressive senescence, a rapid decline in physiological condition resulting in the loss of biomass without external forces such as waves or herbivory[27,29]. While it has been established that senescence increases the rate of particulate detritus shed by giant kelp[30], the impact of senescence on DOC production rates has not been considered.

To address the role of intrinsic and extrinsic factors on DOC production by giant kelp, we performed incubations of giant kelp blades sampled from tagged frond cohorts over several months in the summer and spring periods in the Santa Barbara Channel, CA. We demonstrate that consideration of senescence explains large variability in DOC production by giant kelp. Further, we demonstrate that the senescence-driven DOC production is likely due to the solubilization of standing biomass carbon, rather than by direct exudation. We applied our findings to a novel, large-scale time-series data set of giant kelp canopy biomass and physiology. Our results demonstrate that senescence-driven solubilization drives most of the DOC released from giant kelp to the coastal ocean.

## Results
### Age and seasonally driven changes in kelp physiology and NPP
To better understand how intrinsic and extrinsic factors influence the partitioning of NPP into DOC production by giant kelp, we measured both from kelp sampled during nutrient-deplete (summer) and replete (spring) periods over blade ages of 16–78 days (Supplementary Table 1). The ages of sampled blades in both seasons covered the periods from early to late maturity (16-43 days) through early to late senescence (58-78 days). We observed large, rapid, and non-linear changes in kelp physiological condition, as measured by its Chl:C content, after 50 days of age in both seasons and hereby refer to kelp tissue younger or older than 50 days as "mature" or "senescent", respectively (Fig. 1a). Mature summertime giant kelp C:N (mol:mol) was on average $34.6 \pm 4.1$, three times larger than average mature springtime C:N (mean $\pm 1$ SD $= 11.8 \pm 0.8$). In both the summer and spring cohorts, there was an increase in average tissue C:N with age (Supplementary Table 1). Mature spring kelp had a significantly higher tissue Chl:C content (Welch's *t* test, $t = 5.8$, df $= 13.6$, $p < 0.001$) and significantly lower tissue C:N (Welch's *t* test, $t = -22.5$, df $= 19.0$, $p < 0.001$) than mature summer kelp.

Across all incubations, NPP rates ranged from $-30.3$ to $264.9 \, \mu mol \, C \, g_{DW}^{-1} \, h^{-1}$ (Fig. 1b). As expected, rates of NPP displayed a non-linear response to light, increasing rapidly with exposure to low light levels and saturating at PAR values $> 300 \, \mu mol \, photons \, m^{-2} \, s^{-1}$ (Supplementary fig. 1a). In both seasons, there was a significant linear decrease in maximum photosynthetic rates with age (OLS) regression, Summer: $R^2 = 0.85$, $p < 0.001$, $n = 30$; Spring: $R^2 = 0.44$ $p < 0.001$, $n = 24$, although the spring cohort had a slower rate of decline with age than the summer cohort

(Supplementary fig. 1b). Negative photosynthetic rates reported are apparent respiration rates when PAR was equal to zero.

### Giant kelp DOC release rates
Rates of DOC release ($DOC_{ex}$) by giant kelp blades were influenced by both extrinsic and age-driven intrinsic processes, namely light, NPP, and senescence (Fig. 2). $DOC_{ex}$ ranged from $-1.2$ to $65.3 \, \mu mol \, C \, g_{DW}^{-1} \, h^{-1}$ across all incubations (Supplementary Data 1). Two data points were excluded from our analysis due to accidental physical damage to the kelp tissue by the incubator stir bars, resulting in artificially high $DOC_{ex}$. These data points are included, and their exclusion is discussed in Supplementary fig. 2.

Within each season, there was a significant increase in $DOC_{ex}$ between mature and senescent kelp (Wilcoxon Test, Summer: $W = 189.5$, $p = 8.7e^{-10}$; Spring: $W = 275$, $p = 2.7e^{-5}$). In mature kelp blade incubations, there was a significant linear correlation between rates of NPP and $DOC_{ex}$ (Fig. 2a; Model II; $R^2 = 0.27$; $p = 1.81e^{-7}$, $n = 88$). Percent extracellular release (PER) was calculated as $DOC_{ex} /NPP \times 100\%$ for incubations where NPP > 0. In mature kelp incubations, average PER ($\pm 1$ SD) was 2.7 ($\pm 1.2$) % and 2.3 ($\pm 2.2$) % of NPP, in the spring and summer, respectively. As a test of the overflow hypothesis[22], we compared the relationship between PER and tissue C:N and light intensity. Although we found a significant negative relationship between PER and tissue C:N, opposite to the predictions of the overflow hypothesis, it was a poor predictor variable (Supplementary fig. 3a; Model II, $p = 0.038$, $R^2 = 0.06$, $n = 73$). For example, across a gradient of tissue C:N from 10 to 40 it would only predict a change in PER from 3.0% to 1.6%, a range within one standard deviation of the average PER in both seasons. In addition, PER showed no significant variability with irradiance level (Supplementary fig. 3b, OLS, $p = 0.48$, $R^2 = -0.006$, $n = 73$). In mature kelp incubations, $DOC_{ex}$ continued in the dark (PAR $= 0 \, \mu mol \, photons \, m^{-2} \, s^{-1}$) at an average ($\pm 1$ SD) rate of 0.9 ($\pm 1.0$) $\mu mol \, C \, g_{DW}^{-1} \, h^{-1}$, approximately three times lower than rates in light-saturating conditions ($306–1517 \, \mu mol \, photons \, m^{-2} \, s^{-1}$), which averaged ($\pm 1$ SD) 3.3 ($\pm 2.0$) $\mu mol \, C \, g_{DW}^{-1} \, h^{-1}$. $DOC_{ex}$ by mature kelp was also positively correlated with light intensity (OLS; $R^2 = 0.14$, $p < 0.001$, $n = 88$), but light intensity was a weaker predictor variable than the rate of NPP.

As blades entered the senescent phase, $DOC_{ex}$ became uncoupled from NPP (Fig. 2b) and was not correlated with light intensity (OLS, $R^2 = 0.00$, $p = 0.54$, $n = 70$). This decoupling of $DOC_{ex}$ and NPP with age occurred in both the spring and summer cohorts following the onset of senescence (Fig. 2b, Supplementary Table 1). Notably, $DOC_{ex}$ in the senescent phase often equaled or exceeded simultaneous rates of NPP. These elevated rates continued in the dark, suggesting a continuous, large release of DOC by senescent blades, but were highly variable across all senescent blade incubations (mean $\pm 1$ SD $= 14.0 \pm 14.1 \, \mu mol \, C \, g_{DW}^{-1} \, h^{-1}$). This large variability in senescent kelp $DOC_{ex}$ is, in part, due to the progressive senescence of giant kelp blades as they aged beyond 50 days. We observed that senescent kelp $DOC_{ex}$ rates increased as physiological condition, measured as blade chlorophyll *a* content normalized to the maximum observed in each seasonal cohort, declined (Supplementary fig. 4; Model II, $R^2 = 0.35$, $p < 0.001$, $n = 70$).

### DOC Composition
The total carbohydrates fraction released by giant kelp blades remained a relatively constant proportion of the released DOC in all incubations, averaging $10.3 \pm 4.9\%$; however, the relative molar contribution of some hydrolyzable sugars to the total moles of hydrolyzable sugars released (mole %) was more variable. For example, we observed a significant difference in the mole% of sugars in the kelp exudates between the mature and senescent stages (Fig. 3a, PERMANOVA; $p = 0.001$, $R^2 = 0.14$, $n = 42$, Supplementary fig. 5). These differences were mostly driven by the mole% of fucose and mannuronic acid (Man-URA) which constituted an average of 47% and 34%, respectively of the sugars exuded in the mature and senescent phases, respectively (Fig. 3c, d). On average, fucose comprised 47% and 32% of the carbohydrate monomers from mature and senescent kelp exudates, respectively. Man-URA had the largest change in mole% of all sugars

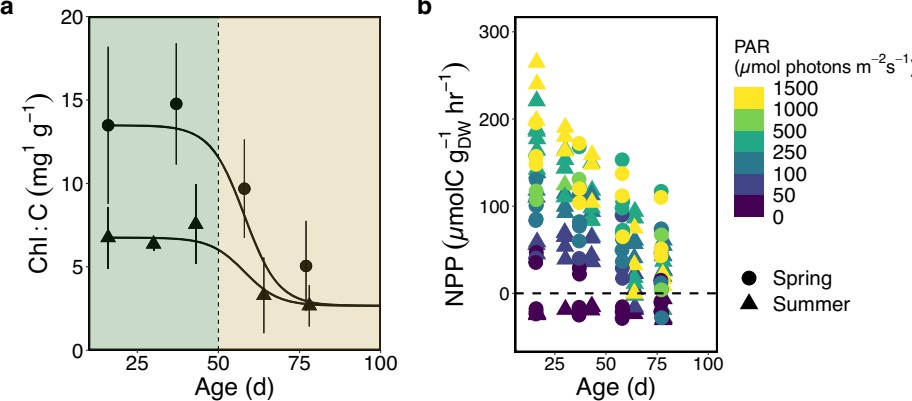

**Fig. 1 | Physiological state and NPP are a function of environmental conditions and age. a** Age-related changes in the tissue chlorophyll *a* to carbon ratio in the spring (circles) and summer (triangles). Shading on either side of 50 days represents the transition between mature (<50 days) and senescent (>50 days) giant kelp. Solid lines are a sigmoidal fit to emphasize the non-linear decline in Chl:C with age. Curves were manually fit assuming a maximum age of 100 days and a minimum

Chl:C equal to the average Chl:C for the summer cohort at 78 days of age. Error bars are ±1 SD from the mean for the six replicate blades sampled for each age. **b** Rate of net primary production (NPP) by giant kelp blades in response to gradients in age and light intensity (photosynthetic photon flux density of PAR). Trends for the spring (circles) and summer (triangles) are shown. The dashed, horizontal line represents the transition between net respiration and net photosynthesis.

**Fig. 2 | Relationship between DOC exudation (DOC$_{ex}$) and NPP across environmental and physiological gradients. a** Rates of DOC$_{ex}$ by mature giant kelp blades (<50 days of age) vs. NPP over a gradient of light levels. The solid line is the significant linear relationship between DOC$_{ex}$ and NPP for mature blades (Model II, $R^2 = 0.27$, $y = 0.015x + 0.96$, $p < 0.001$). **b** The DOC$_{ex}$ vs. NPP relationship across a gradient of blade ages, including mature (<50 days) and senescent (>50 days) kelp blades. The solid black line is the regression line from **a**, and the dashed line is the 1:1 line. Data points to the left of the dashed line are indicative of kelp tissue solubilization to DOC.

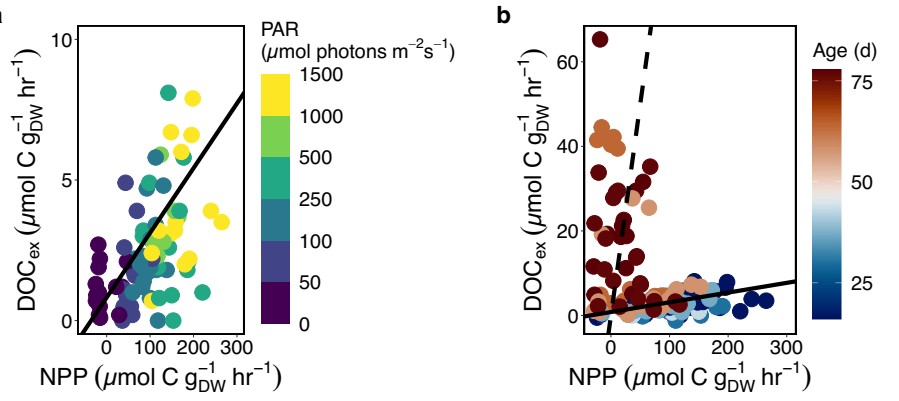

between the mature and senescent phase, increasing 7-fold from an average mole% of 5–34%, respectively (Fig. 3c, d).

### Regional estimates of giant kelp canopy biomass, physiology, and senescence-driven DOC production

Monthly changes in canopy biomass across central and Southern California between 2001–2023 were assessed using Landsat multispectral imagery. At this scale, giant kelp showed a seasonal pattern of growth in the spring, resulting in a peak in biomass in the summer (Fig. 4a). Total giant kelp biomass showed large intra- and interannual variability, ranging from 2 to 371 Gg (1 Gg = 1000 metric tons) of wet biomass in our time series across central and southern California. By tracking daily changes in biomass, we found that the fraction of kelp canopy biomass that was senescent (>50 days old) in our study region followed a seasonal cycle; the senescent portion of canopy biomass was lowest in the spring, increased through the summer and peaked in the fall (Fig. 4b).

We applied our observed dry mass-normalized DOC$_{ex}$ rates to the satellite-derived estimates of giant kelp canopy biomass and physiological state. We used rates from our dark and light-saturating incubations for mature kelp (mean ± 1 SD = 0.9 ± 1.0 and 3.3 ± 2.0 µmol C g$_{DW}^{-1}$ h$^{-1}$, respectively) and given our observation of no relationship between senescent kelp DOC$_{ex}$ and light intensity, we assumed DOC$_{ex}$ from senescent

kelp (mean ± 1 SD = 14.0 ± 14.1 µmol C g$_{DW}^{-1}$ h$^{-1}$) did not follow a 12-hour light/dark cycle. The uncertainty in these rates was accounted for by bootstrap analysis with 100,000 simulations. We generated probability distributions and bootstrap statistics (median ± standard error; 95% confidence intervals) for mature, dark DOC$_{ex}$ (0.7 ± 0.3 µmol C g$_{DW}^{-1}$ h$^{-1}$; 0.2–1.3 µmol C g$_{DW}^{-1}$ h$^{-1}$), mature, light-saturating DOC$_{ex}$ (3.2 ± 0.3 µmol C g$_{DW}^{-1}$ h$^{-1}$; 2.6–3.8 µmol C g$_{DW}^{-1}$ h$^{-1}$), and senescent DOC$_{ex}$ (6.5 ± 2.2 µmol C g$_{DW}^{-1}$ h$^{-1}$; 4.6–12.7 µmol C g$_{DW}^{-1}$ h$^{-1}$). The medians and 95% confidence intervals were extrapolated to monthly estimates of giant kelp canopy biomass and physiology (Fig. 4) across the central and southern California coast, including the California Channel Islands (Fig. 5a). Monthly rates were summed to generate annual DOC production rates for giant kelp between 2001 and 2023 (Fig. 5b). Although we lack measurements from the winter, our sampling covered the full spread of giant kelp physiological condition over an annual cycle (Supplementary Table 1; Supplementary fig. 6).

By applying a binary physiological state (mature or senescent) to our estimates of giant kelp canopy biomass, annual DOC production rates increased on average two-fold compared to when we did not account for senescence (Fig. 5b). Annual DOC production rates for giant kelp averaged (±1 SD) 4.4 ± 1.9 and 2.1 ± 0.9 Gg C yr$^{-1}$, with and without including senescence, respectively. On average, the contribution from senescence-

**Fig. 3 | Changes in giant kelp exudate sugar content between physiological states suggest structural carbohydrates, such as alginate are being solubilized during senescence. a** Principle component (PC) analysis of giant kelp carbohydrate exudate sugar content expressed as molar percentages between mature and senescent phase kelp. Ellipses represent 95% confidence regions between mature (blue circles) and senescent (red triangles) kelp exudates. Arrow lengths represent the strength of the correlation between each individual sugar monomer to the two principal components (PC1 & PC2) shown. Large points in the center of each ellipse are the centroids. Sugar monomer names are overlaid next to arrows. Abbreviations: Glc-URA (glucuronic acid), Gal-URA (galacturonic acid), Man-URA (mannuronic acid). **b** Relationship between rate of DOC production by giant kelp and the mole% of Man-URA in dissolved carbohydrates. Solid line represents the significant Model II regression between the two variables ($y = 89.7 \times x - 6.22$, $R^2 = 0.50$, $p < 0.001$, $n = 42$). Error bars in the $y$ axis are the ±1 standard deviations from the mean for the DOC production rates by a single blade incubated across multiple light levels ($n = 3$). **c** Mole % of fucose and mannuronic acid (Man-URA) in carbohydrates exuded by giant kelp at different ages in the summer and **d** spring. Box and whiskers show the interquartile range, with the median and the variability outside the first and second quartiles, respectively. The $x$ axis is not continuous, and for each discrete age shown on the $x$ axis there is a value for both the mole% of Fucose and Man-URA.

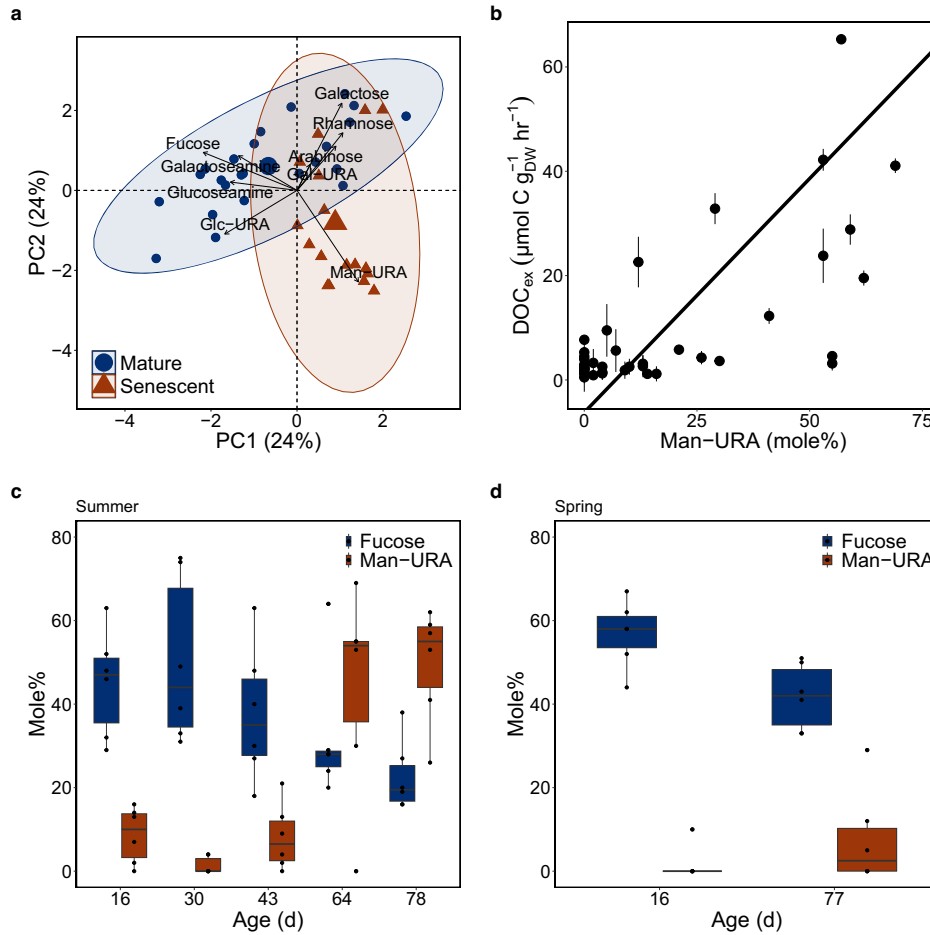

driven DOC release would account for 74 ± 3% of total annual DOC production by giant kelp.

## Discussion

CVEs are recognized for their outsized contribution to carbon storage[31]. However, the role of marine macroalgae in carbon sequestration remains contentious[13,14]. A potential pathway for macroalgae carbon sequestration may be the amount that is exported as DOC[3,12,32], however this is poorly constrained. Current estimates of global DOC production by macroalgae apply rate measurements from short-term incubations, with macroalgae of unknown physiological condition[33], to some measured or assumed standing stock of macroalgal biomass[12]. However, macroalgal biomass varies seasonally and interannually[34,35], and following periods of growth, biomass physiology can change rapidly due to processes such as senescence or nutrient limitation[29,36]. In our study, we demonstrate that knowledge of kelp's physiological condition, in addition to estimates of standing biomass, greatly improves our understanding of DOC production by kelp and its contributions to coastal carbon budgets.

### Seasonal and age-driven changes in physiology and giant kelp senescence

Macroalgae physiology can vary widely across temporal and spatial scales[28,37]. We used tissue C:N and Chl:C ratios as proxies for giant kelp nutrient stress and physiological state, respectively, across seasonal and age-driven gradients. Together, the observed age-dependent decline in photosynthetic rates and Chl:C, and increase in tissue C:N, is consistent with the dynamics of progressive senescence in giant kelp populations, and autotrophs in general[24,29,38]. In both seasons, the increase in tissue C:N began after 50 days,

suggesting that kelp ceases to invest nitrogen resources in blades near the end of their lifespan. Progressive senescence has been studied extensively in terrestrial plants[24,39], however, it has only recently been studied in macroalgae species such as giant kelp[27,38].

The most striking feature of our photosynthetic rate measurements presented was the linear decline in maximum photosynthetic rate with age in both cohorts (Fig. 1b), which has been observed previously for giant kelp[38]. Linear age-related declines in maximum photosynthetic rate are consistent with the predictions of leaf-lifespan theory[40]. This theory posits that leaves, and in the case of giant kelp, blades, seek to maximize their photosynthetic gains against the cost of biosynthesis and maintenance. It predicts that leaf lifespans are shorter when initial photosynthetic rates are high and longer when biosynthesis costs are higher or initial photosynthetic rates are low. Our results are consistent with this theory as we observed a more rapid decline in maximum photosynthetic rates in the summer, when the tissue C:N ratio was highest, and a slower decline in the spring when the tissue C:N ratio was lowest (Supplementary fig. 1b). Of important relevance to this study, we observed that this age-related senescence resulted in a large increase in DOC$_{ex}$ by giant kelp (Fig. 2b).

### DOC$_{ex}$, photosynthetic rate, and light

We observed high variability in hourly DOC$_{ex}$, for the mature and senescent kelp blade incubations, ranging from −1.2 to 8.1 and 0.2 to 65.3 μmol C g$_{DW}$$^{-1}$ h$^{-1}$, respectively (Supplementary Table 1). For mature kelp, this variability was driven by rates of photosynthesis (Fig. 2a), which was a function of both age and light (Fig. 1b). Sampled kelp blades were each incubated across limiting (0–300 μmol photons m$^{-2}$ s$^{-1}$) and light-saturating intensities (300–1517 μmol photons m$^{-2}$ s$^{-1}$) for 2–3 hours, and in mature kelp incubations, DOC$_{ex}$ was linearly correlated to NPP, even

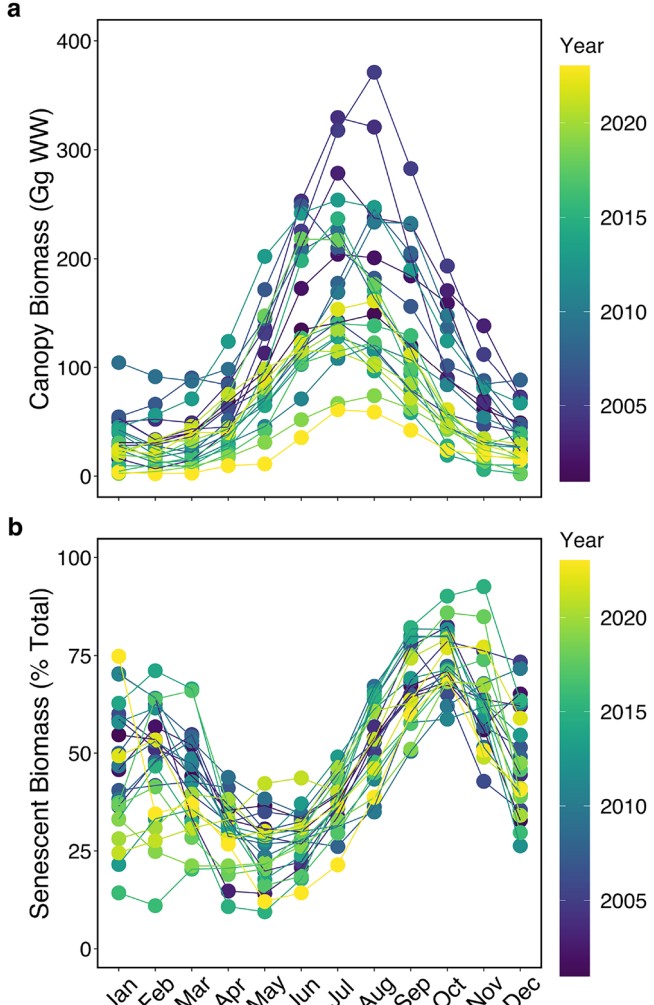

**Fig. 4 | Intra- and interannual variability in giant kelp canopy biomass (in Gg of wet weight) and physiological state estimated from Landsat imagery across the central and southern California region. a** Monthly estimates of giant kelp canopy biomass between 2001–2023 derived from Landsat 7, 8, and 9 multispectral sensors. Note: 1 Gg = 1000 metric tons. **b** Percentage of total monthly biomass in **a** that is senescent (>50 days old). In both panels, interannual variability is shown by the point and line color. A continuous version of this figure is available in the supplemental section (Supplemental fig. 7) in order to see the data by individual years more clearly.

at light intensities higher than the saturating irradiance (Fig. 2a). This indicates that the rate of $DOC_{ex}$ responds rapidly to changes in light but is ultimately constrained by the rate of photosynthesis. This result is consistent with the only other macroalgae study we are aware of that measured simultaneous changes in $DOC_{ex}$ and photosynthesis in response to rapid changes in light[41]. Therefore, models that assume a simple linear relationship with light may overestimate the proportion of NPP released as DOC by non-senescent macroalgae, as $DOC_{ex}$ would continue to increase beyond light intensities where NPP is light-saturated. One such model was used by Reed et al.[9], who estimate that giant kelp releases on average 14% of NPP as DOC annually, higher than our average measured PER (~2–3%). In their study, they did not measure $DOC_{ex}$ and NPP simultaneously, but rather combined mass-normalized $DOC_{ex}$ using a simple linear relationship with light with an existing model of giant kelp NPP. Further, they did not differentiate $DOC_{ex}$ by mature or senescent kelp, which, coupled with a simple linear light-$DOC_{ex}$ relationship, may explain their higher estimated PER.

### DOC exudation mechanisms of mature kelp

One of the main models for DOC exudation by autotrophs, known as the overflow hypothesis[22] predicts that as algae become nutrient stressed, a greater proportion of recently fixed carbon is released as DOC. According to this hypothesis, algae will release photosynthate in greater proportions relative to NPP when light and nutrients are uncoupled. In our study, we observed a nearly 3-fold difference in the tissue C:N of mature blades between the spring and summer (C:N ~ 10 - 40), a difference that spans the long-term observations of giant kelp stoichiometry at our study site (Supplementary Data 1, Supplementary fig. 6). Summertime tissue C:N (~34) values were close to the observed maximum for giant kelp at our study site indicating extreme nitrogen depletion[42,43], while springtime tissue C:N values (~11) were typical for this time of year at our study site (Supplementary fig. 6). However, DOC release rates, as a fraction of NPP, by mature blades remained relatively constant across variable tissue C:N (Supplementary fig. 3a), contrary to the predictions of the overflow hypothesis. A possible explanation for a relatively small and constant percent extracellular release (PER) despite a large range in tissue C:N is the body plan of giant kelp. Giant kelp and several other brown macroalgae contain phloem-like transport networks capable of transporting carbohydrates, such as glucose and mannitol, over a meter per day[44,45]. Unlike phytoplankton, for whom the overflow hypothesis was initially proposed, kelps are multicellular and can transport excess photosynthate to tissue beneath the canopy that may be light-limited. In the interior of a giant kelp forest, light intensity only a few meters beneath the surface can be less than <10 µmol photons $m^{-2}$ $s^{-1}$, several hundred times lower than surface irradiances[38]. Therefore, the release of excess giant kelp photosynthate as DOC by canopy blades at the surface would deprive the biomass below the canopy that relies on this excess photosynthate as a carbon source. In a study of resource translocation of carbon by giant kelp, it was found that canopy blades, like the blades studied here, are important sources of carbon for new frond growth[46]. We hypothesize that DOC release by giant kelp serves an alternative function to energy dissipation and could include the release of DOC for UV protection[47], herbivory deterrence[48], the establishment of their microbiome[49], or drag reduction[50].

### Senescence results in the solubilization of kelp biomass

Senescence is known to play a major role in the spatial distribution and biomass of primary producers[27,39], yet, its role in partitioning biomass between dissolved and particulate detritus is not included in current global estimates of macroalgal biogeochemistry[10,12]. $DOC_{ex}$ rates for senescent kelp blades were considerably higher than observed for mature kelp and were uncoupled from rates of photosynthesis and light intensity (Fig. 2b). $DOC_{ex}$ during senescence increased with the level of physiological decline of the kelp tissue which generally increased with age after the onset of senescence (Supplementary fig. 4). Comparatively, $DOC_{ex}$ rates often exceeded the simultaneous rate of NPP during senescence (Fig. 2b), indicating the loss of previously fixed carbon as DOC through solubilization (i.e., the transformation of particulate organic carbon into DOC), rather than by direct exudation. This apparent solubilization of kelp biomass was supported in our analysis of the dissolved carbohydrates released by giant kelp (Fig. 3a) and the positive relationship between $DOC_{ex}$ and the proportion of mannuronic acid (Man-URA) in released dissolved carbohydrates (Fig. 3b). Man-URA is one of the two acidic sugars (with guluronic acid) in alginate, a carbohydrate that makes up to half of kelp biomass and is a major cell wall polymer[51]. The enrichment of Man-URA in the dissolved exudates, coupled with high $DOC_{ex}$ relative to NPP after 50 days, indicates the solubilization of alginate into the dissolved phase.

Despite the observed solubilization of biomass in kelp older than 50 days, senescent kelp blades were not dead and continued to photosynthesize, albeit at lower rates (Fig. 1b, Supplementary fig. 1b). A possible cause for the progressive solubilization of kelp tissue following the onset of senescence is the growth of epiphytic bacteria, whose hydrolytic enzymes breakdown structural compounds. Kelps contain little cellulose and no lignin, but maintain the structural integrity of their cell walls with a combination of sulfated carbohydrates, such as fucoidan, and carbohydrates rich in acidic sugars, such as alginate[52]. Bacteria are abundant on the surfaces of kelp and prioritize the degradation of alginate over other structural carbohydrates[53,54]. This degradation is performed by bacteria that are

**Fig. 5 | Annual DOC production by giant kelp across central and southern California. a** Average standing giant kelp canopy biomass (in kg of wet weight) in 500 m latitudinal bands between years 2001–2023. **b** Annual DOC production for the region in (**a**) between 2001–2023 with (gold lines) and without (green lines) consideration of senescence. Rates were calculated using satellite-derived canopy biomass and age, with our mass-specific $DOC_{ex}$ rates derived from our incubations. Solid and dashed lines show the rates derived from the median and 95% confidence intervals, respectively, from the uncertainty analysis of our $DOC_{ex}$ rates.

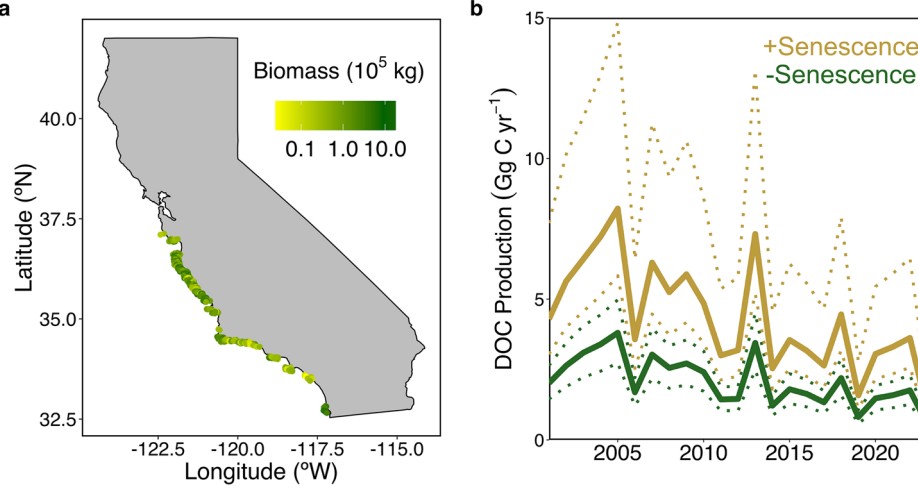

initially rare on the surfaces of the kelp[55], suggesting that as kelp age, microbiome disruption can enhance tissue degradation. Indeed, as part of a complementary study[56] we observed changes in giant kelp's microbiome during senescence, including an increase in the relative abundance of Flavobacteria and Proteobacteria, two groups enriched in alginate-degrading bacteria[57]. Although some bacterial alginate lyase enzymes are tethered to the cell surface (ectoenzymes) to allow efficient scavenging of the hydrolyzed sugars, some bacteria use untethered enzymes (exoenzymes) that can result in the efflux of degradation products like smaller poly- and oligosaccharides[58,59]. The broadcasting of alginate lyase enzymes by epiphytic bacteria may be responsible for the observed solubilization of kelp biomass in our study, ultimately resulting in a pulse of DOC into the surrounding seawater during kelp senescence. This process is well-described in sinking marine particulate organic matter, where bacteria solubilize polymers faster than products can be taken up, resulting in plumes of DOC[60–62]. We propose that solubilization is a major avenue for giant kelp biomass transformation into the marine DOC pool.

Senescence-driven solubilization may also impact the lability of kelp-derived DOC. For example, fucose-rich carbohydrates, such as fucoidan, are structurally complex and recalcitrant to bacterial degradation[63]; whereas carbohydrates released during senescence, such as alginate, are more labile and degraded quickly[53,64]. Therefore, senescence may result in the release of large quantities of relatively labile DOC. However, in this study, we did not measure DOC remineralization and cannot comment on the fraction of giant kelp-derived DOC that is recalcitrant.

## Incorporation of senescence into estimates of kelp forest DOC production

Giant kelp grows year-round; however, growth rates and biomass are linked to changing environmental conditions, such as light and nutrient availability, and intrinsic factors, including senescence[27,34]. As a result, a single giant kelp forest stand can have a wide range of blade ages[29]. We observed that at large scales, giant kelp biomass generally follows a seasonal pattern of rapid growth between the spring and summer, followed by a decline through the fall and winter (Fig. 4a), due to senescence and wave disturbances[27,34]. The fraction of senescent blades peaks in the fall, three months after the peak in giant kelp biomass, where on average 68 ± 10% of the total canopy biomass is senescent (Fig. 4a, b). By incorporating a simple binary age structure into our regional observations of giant kelp canopy biomass (Fig. 5a), we found that senescence-driven solubilization is responsible for, on average, 74% of annual giant kelp DOC production. At the upper range, giant kelp contributes up to 8.2 Gg C yr$^{-1}$ (range = 5.8–14.8 Gg C yr$^{-1}$) as DOC to the coastal ocean in central and southern California (Fig. 5b); a small amount of carbon compared to other sources of DOC to the coastal ocean, such as rivers, the largest of which deliver between 230 and 26,900 Gg C yr$^{-1}$ river$^{-1}$ (global total ~250 Tg C yr$^{-1}$)[65].

Our study covers only one kelp forest species in a single region where giant kelp canopies have been observed from satellite imagery (up to ~50 km$^2$ of giant kelp canopy). This is a small fraction of the total potential kelp forest area globally (potential area ~1.96 million km$^2$)[10]. A simple extrapolation of our maximum regional giant kelp DOC production estimate (8.2 (range = 5.8–14.8) Gg C yr$^{-1}$/50 km$^2$) to this potential area would equal a global kelp forest DOC production rate of 321 (227-580) Tg C yr$^{-1}$. This is about twice as high as estimates of kelp forest detrital particulate organic carbon production (~158–307 Tg C yr$^{-1}$)[10], and is equivalent to global DOC production for all macroalgae, not just kelp forests, estimated by Krause-Jensen & Duarte[12] (330 Tg C yr$^{-1}$). However, it is important to note that this estimate assumes kelp forests occupy all available, habitable space[8,10] and therefore represents an upper limit for global kelp DOC production. This assumption is likely rarely, if ever, met. For example, our observations of giant kelp canopy biomass show large intra- and inter-annual variability for a single region; standing canopy biomass at any given time between 2001–2023 is on average (±1 SD) only 23 ± 19% of the maximum observed biomass in August 2005 (Fig. 4a). Further, kelp forests are declining around the world as a result of anthropogenic forces and marine heatwaves[66,67], making it less likely that kelp forests will reach their maximum potential biomass. Future work should prioritize constraining uncertainties in modeled macroalgae biomass and area using in situ observations and remote sensing as part of multi-annual, year-round studies.

This study demonstrates that consideration of physiology is needed to constrain the pathways and fate of macroalgal-derived carbon in the coastal ocean. While not all macroalgae undergo progressive senescence in the same way as giant kelp, there is evidence for seasonal senescence in year-round surveys of other macroalgae species[35,36,68–70]. For example, pelagic *Sargassum* forms extensive blooms in the western North Atlantic and Caribbean Sea, totaling up to 20,000 Gg of wet biomass[35]. After the bloom peaks in the summer, there is a rapid decline in *Sargassum* biomass between July and December, a similar pattern we observed for giant kelp (Fig. 4a). Additionally, three previous studies[18,20,36], encompassing seven species of macroalgae (*Ascophyllum nodosum, Fucus vesiculosus, Fucus serratus, Saccharina latissima* [formerly *Laminaria saccharina*], *Palmaria palmata* [formerly *Rhodimenia palmata*], *Saccharina japonica, Ecklonia cava*), observed elevated DOC release rates in the summer and fall compared to the rest of the year, suggesting that enhanced DOC production as a result of seasonal senescence may be a common feature of macroalgae. This is important to consider for blue carbon estimates, as it would increase the amount of biomass estimated to be exported as DOC, rather than particulate organic carbon, limiting the downward flux of macroalgal organic carbon necessary for sequestration. Future work should determine whether our observed $DOC_{ex}$ rates and seasonal patterns related to senescence can be generalized to all macroalgae.

## Methods

### Kelp collection and incubations
Giant kelp blades were collected from Mohawk Reef (34.3941° N, 119.7296° W) in Santa Barbara, CA, between August 2023 and June 2024. At each sampling event ($n = 9$), six whole blades were clipped between the pneumatocyst and stipe and transported back to a nearshore laboratory in surface seawater and placed in 10 L acrylic incubation tanks filled with 0.2 µm filtered seawater collected the day before. Incubation tanks were submerged in temperature-controlled water near in situ temperature (Summer: 17–19 °C, Spring: 12–14 °C). Blades were allowed 30 minutes to acclimate to the incubation chambers to prevent sampling of exudation driven by handling. Incubation tanks were outfitted with magnetic stir bars to maintain circulation within the chambers. The six collected kelp blades were incubated at three light levels between 0 and 1517 µmol photons $m^{-2}$ $s^{-1}$ for 2–3 hours (Supplementary Data 1).

### Environmental and physiological variables
Incubation photosynthetically active radiation (PAR) was controlled using a dimmable LED light source (VIPARSPECTRA XS4000, Richmond, CA, USA) and measured with a handheld PAR meter (Phantom PHOTOBIO, Chico, CA, USA). Physiological measurements such as age, tissue stoichiometry, and pigment concentrations were determined by previously established methods[28,38,71]. Age cohorts of giant kelp were established in August 2023 (summer cohort) and April 2024 (spring cohort). Tissue age was measured by tagging up to 100−200 growing fronds 2 m back from their apical meristem with a cable tie around their stipe (blade age ~14 days; based on frond elongation rates of ~14 cm $d^{-1}$ [42,72]). The age cohort sampling began two days after the initial tagging. A single blade was sampled at the tag site from six random fronds every 2–3 weeks until we could no longer find our tagged fronds (up to 78 days). Following incubations, the tissue was rinsed with 10% HCl followed by deionized water to remove any $CaCO_3$ from epibionts and dried at 60 °C for 3 days. Dried tissue was weighed, ground to a fine powder, and analyzed for carbon and nitrogen content using a CE-440 CHN/O/S elemental analyzer (Exeter Analytical, Exeter, UK). Chlorophyll $a$ (Chl$a$) concentrations were measured from a 0.8 $cm^2$ disk excised from the tissue before rinsing and drying. Disks were weighed and sequentially extracted in 4 ml of dimethyl sulfoxide and 5 ml of acetone, methanol, and ultrapure water (3:1:1). The absorbance of the extracts was measured from 350 nm to 800 nm (Shimadzu UV 2401PC, Tokyo, Japan)[28,73]. Chl$a$ concentration was calculated from absorption spectra following Seely et al.[71]. The physiological parameter, Chl:C was measured by dividing the mass of Chl$a$ by the dry mass of carbon for each excised disk.

### Net primary production
NPP was measured as changes in dissolved inorganic carbon (DIC) in the incubation seawater. Samples were collected by overflowing a 125 ml glass serum bottle with incubation seawater and preserved with 120 µl of saturated $HgCl_2$. DIC samples were analyzed by acidifying the sample with 10% $H_3PO_4$ and sparging with $N_2$ for 220 seconds. The resulting $CO_2$ in the gas stream was measured via an automated, non-dispersive infrared inorganic carbon analyzer with an AIRICA $TCO_2$ analyzer (MARIANDA, Kiel, Germany)[74]. The $pCO_2$ peak area was converted to µmol C $L^{-1}$ using a coefficient calculated from a certified reference material (CRM Batch #206 & #216; Dickson Lab, San Diego, CA, USA). CRMs were run every 12 samples to check for analytical stability throughout a given run. The average standard deviation from three CRM technical replicates across each run was 2.9 ± 1.9 µmol C $L^{-1}$. Rates of NPP were calculated as follows:

$$\text{NPP}\left(\mu mol\, C\, g_{DW}\, hr^{-1}\right) = \frac{[DIC]_0 - [DIC]_t * V}{T * m} \quad (1)$$

where $[DIC]_0$ and $[DIC]_t$ are the DIC concentrations (µmol C $L^{-1}$) at the beginning and end of each incubation, respectively. $V$ is the volume of seawater during the incubation, $T$ is the incubation time, and m is the tissue dry weight.

### DOC analyses
DOC analysis was carried out according to Halewood et al.[75]. Briefly, duplicate samples for DOC were collected from the beginning and end of each incubation, filtered through pre-combusted 25 mm GF-75 (nominal pore size of 0.3 µm) into pre-combusted 40 mL EPA vials with PTFE lined caps, and acidified to pH ~2 with 4 N HCl. DOC concentrations were quantified by the high-temperature combustion method using a TOC-V or TOC-L (Shimadzu, Tokyo, Japan) using a four-point glucose standard curve. Each run was also referenced against surface and deep seawater collected from near the Bermuda Atlantic Time-Series study site and calibrated against consensus reference material (Hansell Deep Sea Reference Batch #21, Lot#04–21, Miami, FL, USA), run every 6–8 samples. The precision for the analytical runs had a coefficient of variation of duplicate samples <2% or ±0.6 µM C for this study. DOC exudation rates ($DOC_{ex}$) were calculated as follows:

$$\text{DOC}_{ex}\left(\mu mol\, C\, g_{DW}\, hr^{-1}\right) = \frac{[DOC]_t - [DOC]_0 * V}{T * m} \quad (2)$$

where $[DOC]_t$ and $[DOC]_0$ are the DOC concentrations in µmol C $L^{-1}$ at the end and beginning of each incubation, respectively. $V$ is the volume of seawater during the incubation, T is the incubation time, and m is the tissue dry weight.

### Giant Kelp exudate composition
Kelp-derived DOC was analyzed for its carbohydrate content and specific sugar monomer composition. The sugar content of the exudates was measured using high-performance anion exchange chromatography with pulsed amperometric detection (HPAEC-PAD), following dialysis and eluent gradient protocols specified in Engel & Händel[76]. Briefly, samples were dialyzed using Spectra/Por 7 tubing (1000 Da) against ultrapure water, then hydrolyzed for 20 hours at 100 °C in 0.4 M HCl and neutralized under $N_2$. Samples were run on a DIONEX ICS5000+ (Thermo Fisher Scientific) and separated using a Carbopac PA10 column (4 × 250 mm) with a Carbopac PA10 guard column (4 × 50 mm). Neutral and amino sugars were eluted with 18 mM NaOH and followed by 100 mM NaOH/200 mM Na-Acetate to elute acidic sugars. The system was calibrated using a standard sugar mix containing fucose, rhamnose, arabinose, galactosamine, glucosamine, galactose, glucose, mannose+xylose, galacturonic acid, glucuronic acid, and mannuronic acid (Sigma-Aldrich ≥99%). Linearity of the calibration curves was observed for concentrations ranging from 10 nM–1 µM. Due to the leaching of glucose and mannose+xylose-rich carbohydrates from the Spectra/Por 7 dialysis tubing, these sugars were removed from further analysis.

### Estimates of regional giant kelp canopy biomass, age, and DOC production
To extrapolate our measured giant kelp $DOC_{ex}$ rates to regional scales, we determined giant kelp canopy biomass and age distribution using Landsat 7, 8, and 9 multispectral imagery, focusing on the central and southern California coastline where giant kelp dominates. Between the years 2001–2023, we created a spatial time-series of giant kelp canopy biomass estimates at the native Landsat 30 m pixel resolution[26]. Biomass was then interpolated for each pixel to a monthly time scale using a 'makima' interpolation with the interp1 function in Matlab. There was an average of 23.3 (standard deviation = 3.6) cloud-free views per year for each pixel for this region between 2001 and 2023, allowing for this monthly time series to be created. Further, by interpolating each pixel onto a standard monthly grid, we accounted for the effect of tide and current to minimize the uncertainty in our estimates of canopy biomass and age. From this monthly time series, we resampled to a daily resolution and found the difference in kelp canopy biomass between each date using the diff function in Matlab. Positive changes in kelp biomass were then tracked where the first appearance of biomass increased, given an age of one day, and accounted for until the age of 120 days[27], when canopy biomass was assumed to be completely senesced

and lost. By completing this step for each pixel time series, we estimated the age of the canopy biomass for each month of the time series across the study domain. We then multiplied these fractions by the monthly satellite-derived biomass, yielding the wet weight in kg of biomass for all ages for each month and pixel across the central and southern California coastline. Kelp canopy biomass was converted from wet weight to dry weight using the average dry weight:wet weight ratio of 0.12 measured in our incubations.

Using our incubation-derived, dry mass-normalized DOC production rates (Eq. 2), we estimated annual DOC production along the California coastline. We calculated daily $DOC_{ex}$ for mature kelp assuming a 12-hour light/dark cycle and the mass-normalized $DOC_{ex}$ rates measured in the dark (PAR = 0 μmol photons m$^{-2}$ s$^{-1}$) and light-saturating (PAR > 300 μmol photons m$^{-2}$ s$^{-1}$) incubations. We then calculated daily DOC release from senescent kelp using the estimated amount of senescent biomass and our measured senescent $DOC_{ex}$ rates. To account for the uncertainty in $DOC_{ex}$ rates observed in our incubations, we calculated a probability distribution for each of the parameters from our laboratory incubations. We performed a bootstrap analysis with 100,000 simulations to derive a median and 95% confidence interval for $DOC_{ex}$ rates for mature kelp in the dark and light-saturating conditions, as well as for senescent kelp. Daily rates were then used to derive an annual estimate of giant kelp DOC production along the central and southern California coastline between 2001 and 2023.

### Statistical analysis

To compare means between two independent variables that were approximately normal but did not have equal variances, we used Welch's $t$ test. To compare means between two independent variables that were not normally we used the non-parametric Wilcoxon rank sum test. Model II linear correlation analysis was used to compare the relationships when both variables were assumed to have equal random error (i.e., $DOC_{ex}$, NPP). Ordinary least squares (OLS) regression was used to compare the relationship between variables when one variable was assumed not to have random error (i.e., light, age). To visualize how giant kelp-derived exudate composition changed between mature and senescent blades, we conducted a principal component analysis of scaled molar percentages of individual sugars. Differences in the composition of exudates between maturity and senescence were assessed using permutational multivariate analysis of variances (PERMANOVA).

### Reporting summary

Further information on research design is available in the Nature Portfolio Reporting Summary linked to this article.

### Data availability

Data used for analysis, statistics, and figure generation is available in the Supplemental Data 1 file and at https://github.com/chance-english/Giant_Kelp_DOC. Data for Landsat-derived giant kelp canopy biomass is available at the Santa Barbara Coastal LTER data portal (sbclter.msi.ucsb.edu/data/catalog/).

### Code availability

Code used for data analysis, statistics, and figure generation is available at https://github.com/chance-english/Giant_Kelp_DOC.

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

## Acknowledgements
We thank the invaluable work of members of the Santa Barbara Coastal LTER whose data provided important background for our findings. We thank all Carlson lab members for the valuable discussion on the data. We thank A. Santoro for her constructive comments and discussion of the data and manuscript. We also thank the two anonymous reviewers whose comments improved the manuscript. This project was funded by the Department of Energy's Advanced Research Project Agency-Energy through award DE-AR0001559 to T.W.B., D.A.S., and C.A.C., and the National Science Foundation's Santa Barbara Coastal LTER through award number OCE-1831937 to D.A.S. and C.A.C.

## Author contributions
C.J.E. and C.A.C. conceptualized the research. C.J.E. collected material for and ran the incubations. C.J.E., T.W.B., and K.O. collected data and performed laboratory measurements. C.J.E., T.W.B., K.O., D.A.S., and C.A.C. contributed to interpreting the results and writing/editing the manuscript. C.A.C., T.W.B., and D.A.S. acquired funding for the research.

## Competing interests
The authors declare no competing interests.
