## [Transparent Peer Review file · Communications Biology]

Senescence-driven solubilization of biomass is the main source of kelp-derived dissolved organic carbon to the coastal ocean.

Corresponding Author: Dr Chance English

Version 0:

Reviewer comments:

Reviewer #1

(Remarks to the Author)

This is an important and timely study on the relationship between kelp blade physiology, net primary productivity, and dissolved organic carbon (DOC) release rates. The authors found that giant kelp DOC release is primarily controlled by blade senescence by tracking DOC release from tagged cohorts of giant kelp blades to incorporate blade tissue age. They found that DOC release was positively related to net primary productivity before 50 days, but after 50 days, blades undergoing senescence had elevated DOC release that was uncoupled from NPP or environmental variables (light intensity). It is very important to understand the relationship between NPP and DOC release, as >50% of carbon exported from kelp forests is estimated to be in the form of DOC, and quantifying marine CO₂ removal and sequestration by kelp forests is critical to understand their role in the global carbon cycle. I have a few critical methodological questions and comments for the authors to address (below), as well as a few suggestions to improve the manuscript.

Specific comments are below:

Line 107: What was the temperature of the seawater? If not flow-through, were chillers used to maintain cool temps?

Line 110: Missing the word "at"

Line 110: What are the 3 light levels? This is a very wide range, please be more specific

Line 67 & Line 603 – References go from #22 to #24, #23 is missing!

Lines 195-202: From what I understand, you calculated positive biomass as the first appearance of biomass increase for a given pixel sampled across days. Could you please address the possibility of false positive (or negative) change in biomass due to tide height and wave impacts on kelp biomass at the surface from Landsat images taken across time?

Line 204-205: Since cohorts were established in summer & spring, lasting up to 78 days, you did not quantify DOC release in the winter. How can you scale up to annual DOC production from only spring & summer rates? Did you account for possible winter DOC release differences?

Line 234, 236: Is there a reason that you did not track blades past 78 days, if the average age of a giant kelp blade at senescence is 100-120 days (Line 79)? Were these blades in late senescence, if they are only 70% through the blade lifespan (78/110 days)?

Line 298: Please define "mole% of sugars"

Line 430: I would argue the relationship between senescence and DOC release has been studied before – see Carlson et al. 2024 "Kelp dissolved organic carbon release is seasonal and annually enhanced during senescence"

DOC composition: The Discussion section could summarize the DOC composition results, including the composition of

DOC from healthy / mature kelp blades (for example, fucose is not mentioned in the discussion). Also, it would be helpful to present a supplemental figure breaking down the relative proportion of different DOC components in different treatments or seasons, as the total DOC composition is only represented in Fig. 3a, and it does not provide any detail on relative abundance of each component of DOC.

Figure 1: Is there a way to make Fig. 1b less crowded, perhaps by elongating the x axis? It is difficult to see the difference between spring & summer with the overlapping data points.

Figure 3, Line 773-774: please clarify Fig. 3 c-d, at the moment there is no legend for Fig.3c. Clarify in the text what is being represented by the different colors. Why are there only 2 blade age categories in Fig. 3d?

Figure 4: Could you please make the "year" figure legend discrete and not continuous? It would be nice to see the canopy biomass across specific years, for example the marine heatwave of 2014-2016.

Reviewer #2

(Remarks to the Author)

This manuscript describes for the first time the effect of frond senescence on the exudation of dissolved organic carbon by natural populations of kelps, here giant kelp.

The experimental work elegantly made use of the sequential increase in age of kelp fronds in each individual and made a large and consistent work in incubating and measuring various parameters using fronds from two cohorts.

The relevance of the findings is put in perspective by the final part of the work, where the DOC release by a natural kelp population is modelled based on a 20+ time series of kelp forest biomass, with or without taking senescence into consideration.

The manuscript is novel, convincing, and well written. Conclusions are balanced and the implications are large.

Congratulations on a well-planned, well written and very pertinent study and paper!

My recommendation is publication with minor revisions:

General comments:

• Abbreviations/units:

o When mentioning PAR, I would say that the unit should be $\mu\text{mol photons m}^{-2} \text{s}^{-1}$, but 'photons' is consequently absent in this unit throughout the paper, incl. figure axis titles and legends

• Discussion: I would suggest commenting on the lability of the different DOC molecules released over the season, as this is of importance for the potential of sequestration of the C

• Reference list

o Species names should be in italics

• Supplemental material:

o In two figures the kelp tissue C:N ratio is described, and in Supplemental figure 5, it is described that the ratio is calculated on a gram:gram basis. Commonly this ratio is stoichiometric and thus on a mole:mole basis. I recommend that this is changed, to ease comparison to other literature

Specific comments:

• Ll. 127-132: I assume that the C content of the discs from which the Chl a/C ratio of the fronds was calculated was assessed by using the average dry weight to wet weight ratio of the fronds, that were rinsed with HCl prior to drying and estimation of DW. Would there be any potential bias in that the older the fronds from which the discs were cut, the more epibionts they would contain – which could erroneously increase their DM, and here through their C content as relative to their Chl a content – and in this way 'inflate' the assumption of the declining Chl a/C ratio as fronds age? Please, comment on this, as the Chl a/C ratio is fundamental in the paper.

• Ll. 359-362: The aging of another kelp, *Saccharina latissima*, is described in Nielsen et al, 2014, including the impact on the CN content of the fronds (<https://link.springer.com/article/10.1007/s00227-014-2482-y>)

• L. 514: You mention here the species '*Laminaria saccharina*' the current name of this species, however, is *Saccharina latissima* since 2006 (<https://onlinelibrary.wiley.com/doi/10.1111/j.1529-8817.2006.00204.x>). I suggest writing instead '*Saccharina latissima* (former *Laminaria saccharina*)'

• L. 722, a space is missing between 'the' and 'Ecklonia'

• Line 745: In 'Rates of by mature giant kelp blades', 'DOC exudation' is missing between of and by

• Supplemental figure 1, panel B. There is no description of what the two colors, black and red, symbolize.

• L. 860: It is described here that the C:N ratio is calculated on a gram:gram basis – see general comments.

• Supplemental Table 1 – again, the C:N ratio – is it based on gram:gram? See general comments

Version 1:

Reviewer comments:

Reviewer #1

(Remarks to the Author)

Thank you for the thorough edits, the additional methodological details, and the new supplementary figures. The authors' revisions are satisfying and have greatly improved the manuscript!

One requested addition to the methods (line 130): Please explain how individual blades or cohorts of blades were tagged - currently, the methods only describes when the blades were tagged, but not how. Describing the tagging method would be useful for future studies.

Reviewer #2

(Remarks to the Author)

Dear authors,

thanks for a constructive approach to reviewers comments and congratulations on a very good piece of work and subsequent paper.

Reviewer Comments	Author Response
Reviewer # 1	
This is an important and timely study on the relationship between kelp blade physiology, net primary productivity, and dissolved organic carbon (DOC) release rates. The authors found that giant kelp DOC release is primarily controlled by blade senescence by tracking DOC release from tagged cohorts of giant kelp blades to incorporate blade tissue age. They found that DOC release was positively related to net primary productivity before 50 days, but after 50 days, blades undergoing senescence had elevated DOC release that was uncoupled from NPP or environmental variables (light intensity). It is very important to understand the relationship between NPP and DOC release, as >50% of carbon exported from kelp forests is estimated to be in the form of DOC, and quantifying marine CO₂ removal and sequestration by kelp forests is critical to understand their role in the global carbon cycle. I have a few critical methodological questions and comments for the authors to address (below), as well as a few suggestions to improve the manuscript.	We thank the reviewer for their constructive comments. We have reviewed and incorporated changes or clarified methodology where specified.
Line 107: What was the temperature of the seawater? If not flow-through, were chillers used to maintain cool temps?	We have added a note that the incubations tanks were submerged in temperature-controlled DI water bath and included the incubation temperature range for each season.
Line 110: Missing the word “at”	Corrected
Line 110: What are the 3 light levels? This is a very wide range, please be more specific	The three exact light levels for each blade varied between incubations but each blade was incubated across light limiting and light saturating conditions. The exact levels for each incubation are now shown in a new supplemental table 1 (excel file).
Line 67 & Line 603 – References go from #22 to #24, #23 is missing!	Thank you for catching this. We have added the citation to the bibliography.
Lines 195-202: From what I understand, you calculated positive biomass as the first appearance of biomass increase for a given pixel sampled across	During the time period of this study (2001 – 2023) there was an average of 23.3 clear views of each 30x30m

days. Could you please address the possibility of false positive (or negative) change in biomass due to tide height and wave impacts on kelp biomass at the surface from Landsat images taken across time?

Landsat pixel per year. We chose to limit our study to this time period as there were two Landsat satellites in orbit in almost every year, providing multiple clear images of coastline every month. We then interpolated each pixel onto a standard monthly grid to account for the effect of tide and current within each month and minimize the uncertainty in our estimates of canopy biomass and age. To illustrate this (the figure in the row below), we produced a timeseries of a single Landsat pixel between 2011 to 2024. Each blue asterisk shows a clear view of the pixel from the satellite imagery. The red lines connect each clear view, and some noise is evident (likely due to tides and currents). Once the timeseries is interpolated onto a monthly grid (black line), the variability is generally accounted for.

We have added a sentence to the manuscript briefly **(Lines 210-212)** describing our consideration of tides and currents.

Line 204-205: Since cohorts were established in summer & spring, lasting up to 78 days, you did not quantify DOC release in the winter. How can you

We acknowledge our lack of sampling in the Winter; however, we note that our spring and summer sampling

scale up to annual DOC production from only spring & summer rates? Did you account for possible winter DOC release differences?	covered the full range of giant kelp physiology at our site over an annual cycle (C:N ratios [compare supplemental table 2 with supplemental figure 6]). We found that spring and summer DOC exudation rates were not significantly different from one another (Wilcoxon Test, $p = 0.68$), despite a large gradient in tissue C:N. Additionally, because standing biomass in the winter is the lowest in any given year (Figure 4a) contributions from winter DOC release is likely the smallest for the year. But we have added the lack of winter sampling as a note (Line 363 - 366). We also note that Reed et al., 2015 (ref # 11) measured DOC release by giant kelp across a full seasonal cycle and found no evidence of a seasonality on DOC release rates on a per biomass basis.
Line 234, 236: Is there a reason that you did not track blades past 78 days, if the average age of a giant kelp blade at senescence is 100-120 days (Line 79)? Were these blades in late senescence, if they are only 70% through the blade lifespan (78/110 days)?	We have added that we intended to track our fronds as long as possible (lines 132-134) but that 78 days was the longest we were able to achieve. Blade senescence begins when blades reach ~50 days of age. The average age of a giant kelp blade when it is completely lost is ~100 – 120 days. So, for at least 50 days giant kelp undergoes progressive senescence. During this time there is a rapid increase in frond loss rates. We only tagged 100-200 fronds out of likely several thousand at Mohawk Reef, therefore our tagged fronds were either lost or became too difficult to find in a dense kelp canopy after 78 days.

	Many of the blades sampled had no or very little photosynthetic activity relative to respiration by 77 - 78 days (Supplemental Figure 1b), and DOC release in these blades was dominated by structural carbohydrates like alginate, so we argue that we did observe blades in late senescence, although not all blades were equally senescent by this time.
Line 298: Please define “mole% of sugars”	This is now defined in the sentence previous to its usage here (Lines 324 – 325).
Line 430: I would argue the relationship between senescence and DOC release has been studied before – see Carlson et al. 2024 “Kelp dissolved organic carbon release is seasonal and annually enhanced during senescence”	We have updated this line to more specifically state that it has not been considered in current estimates of macroalgal biogeochemistry. We also include the Carlson et al., 2024 reference.
DOC composition: The Discussion section could summarize the DOC composition results, including the composition of DOC from healthy / mature kelp blades (for example, fucose is not mentioned in the discussion). Also, it would be helpful to present a supplemental figure breaking down the relative proportion of different DOC components in different treatments or seasons, as the total DOC composition is only represented in Fig. 3a, and it does not provide any detail on relative abundance of each component of DOC.	I have added a brief discussion on the relevance of a transition from fucose-rich carbohydrates (such as fucoidan) to mannuronic acid-rich carbohydrates (such as alginate) in lines 514 - 520. I have added the mole% of all sugars for each individual sample as supplemental figure 5.
Figure 1: Is there a way to make Fig. 1b less crowded, perhaps by elongating the x axis? It is difficult to see the difference between spring & summer with the overlapping data points.	I have made the points smaller and widened the x-axis to make seeing the overlapping points easier to see.
Figure 3, Line 773-774: please clarify Fig. 3 c-d, at the moment there is no legend for Fig.3c. Clarify in the text what is being represented by the different colors. Why are there only 2 blade age categories in Fig. 3d?	The legends have been included in both panels c and d and the different color boxplots are described in the caption. For the spring cohort we had limited access to the instrument used to measure the sugar monomers and

	could only measure two age timepoints for the spring cohort. We chose to measure the youngest (16 d) and oldest (77 d) tissue ages in this cohort as we would expect to see the largest change in the sugar content between them. Panel d shows the increase in the relative contribution of mannuronic acid (alginate indicator) following senescence (> 50 days).
Figure 4: Could you please make the “year” figure legend discrete and not continuous? It would be nice to see the canopy biomass across specific years, for example the marine heatwave of 2014-2016.	We prefer to keep the scale continuous as the main purpose of the figure is to show the repeatability in peak biomass in the summer, the large inter and intra-annual variability in biomass, and the peak in senescence in the fall. However, we understand the interest is seeing specific years such as during the “blob” marine heatwave. We have added a continuous version of figure 4 as supplemental figure 7, and direct interested readers to it in the figure 4 caption.
Reviewer # 2	
This manuscript describes for the first time the effect of frond senescence on the exudation of dissolved organic carbon by natural populations of kelps, here giant kelp. The experimental work elegantly made use of the sequential increase in age of kelp fronds in each individual and made a large and consistent work in incubating and measuring various parameters using fronds from two cohorts. The relevance of the findings is put in perspective by the final part of the work, where the DOC release by a natural kelp population is modelled based on a 20+ time series of kelp forest biomass, with or without taking senescence into consideration. The manuscript is novel, convincing, and well written. Conclusions are balanced and the implications are large.	Thank you for your comments and suggestions, we have incorporated them into the manuscript as suggested or in some form.

Congratulations on a well-planned, well written and very pertinent study and paper!	
 • Abbreviations/units:  o When mentioning PAR, I would say that the unit should be $\mu\text{mol photons m}^{-2} \text{ s}^{-1}$, but ‘photons’ is consequently absent in this unit throughout the paper, incl. figure axis titles and legends 	I have added “photons” to the PAR legends and indicated in the methods we are measuring the photosynthetic photon flux density.
 • Discussion: I would suggest commenting on the lability of the different DOC molecules released over the season, as this is of importance for the potential of sequestration of the C 	We have added a brief discussion on the impact the observed changes in exudate composition may have on DOC lability lines 514 – 520, however the bioavailability of the exuded DOC was not evaluated in this study.
 • Reference list  o Species names should be in italics 	This has been fixed.
 • Supplemental material:  o In two figures the kelp tissue C:N ratio is described, and in Supplemental figure 5, it is described that the ratio is calculated on a gram:gram basis. Commonly this ratio is stoichiometric and thus on a mole:mole basis. I recommend that this is changed, to ease comparison to other literature 	We have converted all C:N values to molar values.
 • Ll. 127-132: I assume that the C content of the discs from which the Chl a/C ratio of the fronds was calculated was assessed by using the average dry weight to wet weight ratio of the fronds, that were rinsed with HCl prior to drying and estimation of DW. Would there be any potential bias in that the older the fronds from which the discs were cut, the more epibionts they would contain – which could erroneously increase their DM, and here through their C content as relative to their Chl a content – and in this way ‘inflate’ the assumption of the declining Chl a/C ratio as fronds age? Please, comment on this, as the Chl a/C ratio is fundamental in the paper. 	We measured the wet:dry weight of each individual blade and each blade’s Chl:C ratio is derived from its individual wet weight to dry weight ratio, not the average for all blades. Therefore, each Chl:C value accounts for any variability in the wet weight to dry weight between blade. The Chl:C parameter for each individual blade was calculated using specific measurements of wet mass, dry mass, Chl a content, and carbon content of each blade. This data is now available in supplementary table 1 (as an excel sheet)

 • Ll. 359-362: The aging of another kelp, Saccharina latissima, is described in Nielsen et al, 2014, including the impact on the CN content of the fronds (https://link.springer.com/article/10.1007/s00227-014-2482-y) 	Thank you for the reference. I have added it in our conclusion section (Line 565).
 • L. 514: You mention here the species ‘Laminaria saccharina’ the current name of this species, however, is Saccharina latissima since 2006 (https://onlinelibrary.wiley.com/doi/10.1111/j.1529-8817.2006.00204.x). I suggest writing instead ‘Saccharina latissima (former Laminaria saccharina)’ 	Thank you, this has been edited.
 • L. 722, a space is missing between ‘the’ and ‘Ecklonia’ 	This has been fixed.
 • Line 745: In ‘Rates of by mature giant kelp blades’, ‘DOC exudation’ is missing between of and by 	Thank you, this has been fixed.
 • Supplemental figure 1, panel B. There is no description of what the two colors, black and red, symbolize. 	The meaning of the colors has been added to the figure description.
 • L. 860: It is described here that the C:N ratio is calculated on a gram:gram basis – se general comments. 	We have converted these values to mol:mol.
 • Supplemental Table 1 – again, the C:N ratio – is it based on gram:gram? See general comments 	We have converted all C:N ratios to mol:mol